# Nested DWT–Based CNN Architecture for Monocular Depth Estimation

**DOI:** 10.3390/s23063066

**Published:** 2023-03-13

**Authors:** Sandip Paul, Deepak Mishra, Senthil Kumar Marimuthu

**Affiliations:** 1Indian Institute of Space Science and Technology, Trivandrum 695547, Kerela, India; 2Space Applications Centre, Ahmedabad 380016, Gujrat, India

**Keywords:** depth–map, discrete wavelets, nested wavelet net, loss function, training, evaluation

## Abstract

Applications such as medical diagnosis, navigation, robotics, etc., require 3D images. Recently, deep learning networks have been extensively applied to estimate depth. Depth prediction from 2D images poses a problem that is both ill–posed and non–linear. Such networks are computationally and time–wise expensive as they have dense configurations. Further, the network performance depends on the trained model configuration, the loss functions used, and the dataset applied for training. We propose a moderately dense encoder–decoder network based on discrete wavelet decomposition and trainable coefficients (LL, LH, HL, HH). Our Nested Wavelet–Net (NDWTN) preserves the high–frequency information that is otherwise lost during the downsampling process in the encoder. Furthermore, we study the effect of activation functions, batch normalization, convolution layers, skip, etc., in our models. The network is trained with NYU datasets. Our network trains faster with good results.

## 1. Introduction

Information on the depth is useful for applications related to satellite remote sensing, navigating robots, autonomous landing, animal gesture identification, creation of 3D models, etc. Active 3D imaging systems, such as LIDAR, RADAR, SONAR, etc., rely on high–power sources and reflected echoes to build depth maps. Modern mobile platforms prefer minimal resources, and, hence, there is an opportunity for simple 2D visible/infrared cameras here. These cameras are simple, readily available, and use low power. Depth estimation using single 2D images has attracted many research scholars and a trend exists for this method.

Earlier methods of determining monocular depth were variations of detectors, such as focus [1], defocus [2], and apertures [3]. These methods avoided correspondence issues as with stereo methods but required a stack of images and had to address contrast (aperture variation) or magnification (focus variation) issues. Others have recovered depth from a single image using blur cue [4]. These methods have poor performance on homogeneous surfaces and zones with poor contrast. Recently, CNN–based methods were successful in training models to estimate high-resolution depth images from a single image [5]. These networks solve an ill–posed problem after training. The network model architecture consists of customized building blocks, such as convolution layers, pooling functions, activation layers, and expansion layers [6,7,8]. The state–of–the–art CNN use supervised or self–supervised training strategies [6,9,10,11,12,13]. Supervised training [5] requires a labeled depth image for the network to converge. Self–supervision uses stereo image pairs [14,15] monocular video and exploits 3D geometry with image reconstruction or camera pose estimates to estimate depth without labeled data. Recent researchers focused on transformers and attention mechanisms to preserve details of depth image [16]. Authors have implemented innovative loss functions with left–right disparity consistency loss [15], photo–metric loss [17], and symmetry loss [18]. Modern researchers also fused light fields information with photo–metric stereo [19] or used a pair of surface orientation maps (surface normals) from photo–metric stereo [20,21,22,23] to derive accurate depth images.

Image information consists of global features, such as structure, texture, semantic information, and local features, such as edges, noise, etc. After transforming an image into the frequency domain, low frequencies represent the global features, while high frequencies represent the local features. Edges are high–frequency components that represent contours, local regions, and local features in an image [24]. Edges are created by neighboring pixels with significantly different intensities. Networks apply averaging function during pooling or down–sampling, reducing these high–frequency components and thus leading to blurred and jagged edges. This degrades object definitions and merges features with the background which results in errors during depth estimation [25].

Previous works have utilized edge–enhancement methods to improve the depth map accuracy [26,27,28]. Alternately, Discrete Wavelet Transforms (DWT) convert spatial and contrast domain images into the frequency domain and separate low–frequency and high–frequency components. This happens along with down–sampling and avoids artifacts [29,30,31,32]. As DWT provides three different high–frequency coefficients, training can enhance edges while noise can be reduced by learning the required coefficients. The high–quality image can be reconstructed with Inverse DWT (IWT) up–sampling after training.

Most networks for depth estimates are based on DenseNet, ResNet, VGG, etc., which are time and computation intensive. Lately, UNet–like architectures have been used for depth estimation [33,34] for faster learning and implementation in less computationally intensive systems. The skip connections here inherently lead to boundary preservation of depth maps. Earlier these were used for medical analysis and Semantic applications. Ref. [35] used DWT–based down–sampling and up–sampling blocks in UNet–like models.

Our literature survey indicated that monocular depth estimation using lightweight UNet–like networks is under–explored. As the inclusion of wavelets in a network is useful, we propose a moderately dense network using DWT for depth estimates. The application of DWT preserves the detailed features compared to the present UNet and UNet++. This paper discusses our proposed network architecture, variants of this network, performance with the public datasets, ablation studies, and results of experiments carried out. Our main contributions are

A nested DWT–based CNN architecture is proposed for monocular depth estimation,Dense Skip functions are implemented with attention function so as to improve the learning of local features,Dense convolution blocks are incorporated for higher feature extraction and learning.

## 2. Wavelets

Wavelets are orthogonal and rapidly decaying functions of the Wavelet Transform family and provide frequency and location information at lower scales. This 2D DWT disintegrates an image into four components, a low–frequency coefficient map (*LL*) and three high–frequency coefficient maps (*LH*, *HL*, and *HH*). These four coefficient maps have half the resolution of an input image. For the image *Y*, the *DWT* is:(1)LL,LH,HL,HH=DWT(Y)

DWT provides robustness to noise and improves accuracy in deep learning [36]. Another advantage of using DWT is the readily available high–frequency representation of edges. Further, we can learn the coefficients of LH, HL, or HH components to improve the features. The LL map can further be decomposed iteratively to get multi–scale coefficient map sets as shown in Figure 1. In our experimentation, we use a scale of 1. Higher scales are possible, however, we are replacing the pooling operation which reduces the feature map size by two.

The DWT is an invertible transformer. The inverse DWT is IWT which converts the four frequency components back to the original image (twice the resolution of the coefficient maps) without any loss of feature definitions. Both Haar and Daubechies wavelets have been used by researchers. We use the Daubechies wavelet with four vanishing moments (db4) for all models. We also experiment with a Haar wavelet on the best model. Haar wavelet has the simplest basis. We follow the DWT implementation from [37].

## 3. Nested DWT Net Architecture

Deep neural networks train models in a systematic way to extract required image features. Most networks for depth estimates are based on DenseNet, ResNet, VGG, etc., and are time and computationally intensive. Researchers are studying less complex networks for small systems with faster learning. The simplest fully convolutional network is UNet [38] developed primarily for abnormality localization in biomedical images. Here, in this encoder–decoder model, the encoder has multiple blocks. Each block has a stack of convolution operators with the last of the stack feeding a max pooling operator (down–sampling). This sub–block extracts the image context and reduces the image resolution by half. The decoder has an equal number of blocks. Each block has an up–sampling operation that expands the size of feature maps by two. This passes through convolution operations. Skip connections, taken from the corresponding encoder block, are fed to the decoder block to enhance the output predictions. The decoder adds localization information to the input context information. The final output has a similar resolution to the image taken for prediction. UNet provides detailed segmentation maps using limited training datasets.

UNet was further improved by [39] by adding soft attention. Using attention, the network learns to focus on relevant zones with low computational complexity. Attention to the skip connections reduces redundant image features and improves prediction accuracy. Residual UNet was developed [40] to overcome accuracy degradation and ease the training of networks. Refs. [31,41,42,43,44,45] proposed wavelet transform to extract sharp edge and strong object boundaries for image dehazing, enhancing the low–light image, reducing artifacts, image segmentation, and depth map estimation. Here, they replace the down–sampling layer with a DWT and the up–sampling layer with an IWT. Ref. [30] used multiple wavelet–based transforms for down–sampling. Recently, Refs. [31,41,46] studied wavelet–based loss function to improve structural details while [32] proposed learning sparse wavelet coefficients to estimate depth maps. Researchers have also replaced the UNET encoder backbone with dense pretrained image networks, such as ResNet, DenseNet [7], etc., and tuned the decoder accordingly to estimate depth. However, these deep networks need computationally intensive resources. UNET++ was defined by [47] and uses (a) convolution operations on skip paths, (b) a series of nested skip paths (dense), and (c) deep supervision. The prediction of this model was improved by reducing the gaps in encoder–decoder feature maps at the sub–block levels. This also made learning easier. The published performance is better compared to UNet and wide UNet. To date, UNET++ has been mostly used for medical analysis and Semantic applications [48]. We realize that this is a less researched area for estimating depth maps. We study UNet and UNet++ to propose a moderately dense network using Discrete wavelets to preserve depth map details.

Our network architecture, Nested Discrete Waveform Transform Net or NDWTN for depth estimation uses an encoder, multi–scale decoders, and skip paths. We replace the down–sampling and up–sampling layers with wavelet transforms. All the coefficients of the wavelet transform (LL, LH, HL, HH) are trainable so as to preserve the details. We also apply nested dense skip paths and convolution, such as UNET++. We also evaluate variants of NDWTN by implementing (a) attention in the skip path, (b) residual blocks in place of convolution blocks, (c) batch normalization layers, and (d) different activation layers. Our network architecture is shown in Figure 2. The structure is similar to UNet++ and has a single encoder path and multiple decoder paths of different scales, all connected through dense skip connections. These skip connections enable nested networks which reduces the semantic gap and provides deep supervision of the output. NDWTN has four scales having a UNet structure. The networks and scales are indicated with yellow, blue, green, and pink colors. Each decoder has independent outputs which are connected to the final output through skip connections.

An encoder down–sampling block is shown in Figure 3 and consists of a learnable convolution block and a DWT layer. This block can have residual convolution as indicated by ‘+’ sign. DWT layer downsamples the input. This block provides input to the skip layer and the succeeding block. A typical convolution block includes convolution, activation, and batch normalization layers. The activation function can be either ReLU (Rectified Linear Unit) or Leaky ReLU. The number of each layer can be increased to improve network density Figure 4. Convolution blocks can be replaced with residual convolution blocks. Residual blocks have additional skip paths. These skip paths are taken from the output of the first convolution operator in the block and then added to the final output of the block. A typical residual block is shown in Figure 5. Each down–sampling block feeds the next block and the skip paths. There are five down–sampling blocks.

A decoder up–sampling block is shown in Figure 6 and consists of an IWT layer for up–sampling and a learnable convolution block. The convolution block is similar to the encoder and can be replaced with a residual block. Two inputs are received (a) from the preceding block and (b) from the skip function. The input from the skip layer is concatenated. The number of convolutions, batch normalization, and activation layers in the convolution block is variable. A typical full decoder block is shown in Figure 7. Each up–sampling block feeds the next block. There are five up–sampling blocks.

The supervised training attempts to predict pixel–wise depth from models by minimizing the regression loss with the help of ground truth images. In the network, the down–sampling and convolution layers of the encoder reduce the information details of these input images. The feature maps arising from initial encoders have higher resolution but lack global information when compared with the final encoders. Our encoder–to–decoder skip connections improve these detail preservation by concatenating high–resolution local features from initial encoders with the decoders global contextual features. Each skip connection responds to the encoder features energy using a gating signal to generate an attention grid that preserves the higher local features.

Typically, the skip layer feeds the encoder output to the decoder. The skip function improves the feature details while up–sampling in decoders. The skip paths go to every decoder horizontally (same scale). This provides nested dense skip functions totaling 10. We also add an attention gate inside the skip layer. This skip layer takes two inputs, one from the preceding encoder block (higher scale) and one from the horizontal encoder block (same scale), and provides input to the attention gate. A strided–convolution operation on the lower input lets us match the feature dimensions. Summing these two vectors results in higher aligned weights and small unaligned weights. The resultant vector is fed to a ReLU function followed by a convolution to reduce the feature dimensions to 1. This vector is finally scaled within [0, 1] using the Sigmoid activation function. This stage gives the attention coefficients. The attention coefficients are upsampled to the dimensions of the lower scale and multiplied with the lower scale input. This input is given to the decoder block. Figure 8 represents the attention block.

Finally, the decoder output layer is represented by Figure 9. The block is the same for all four decoder outputs. The block has the Sigmoid function as the activation layer to provide the final prediction statistics. Though we can use the scaled four outputs independently, we feed three outputs to the final fourth output via nested skip layers to have the best prediction accuracy. We made variants of our network based on convolution blocks, residual convolution blocks, and attention features. These architecture variants are designated as:NDWT: Basic NDWTN;NADWT: NDWTN with attention on skip paths;NRDWT: NDWTN with residual blocks;NARDWT: NRDWT with attention on skip paths.
Figure 9Output block with Sigmoid activation layer.
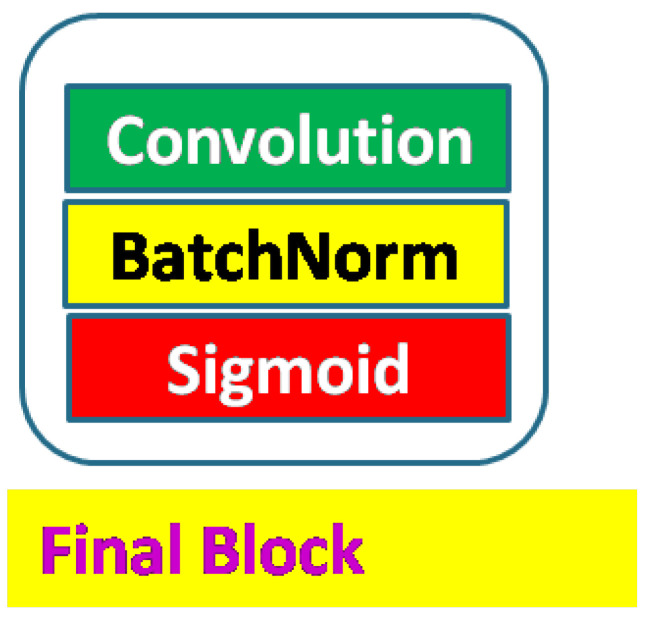


Table 1 compares our model with published models. The trainable parameters in our models are higher than most UNet types. The non–trainable parameters are less than DenseNet backbone UNet [7].

The input RGB image resolution used is 240 × 320 × 3 pixels and the labeled data resolution is 240 × 320 pixels. The estimated depth map resolution is also 240 × 320 pixels. These dimensions are arbitrarily chosen.

## 4. Loss Function

A loss function reduces the training loss and estimates depths comparable to the reference labeled data by converging the model, through gradient descent, during training. Hence, an optimum formulation is crucial for good performance and faster training. The convergence is helped by error functions such as the Mean Absolute Error (MAE) or the L1 loss function. This loss is robust to outliers as it has a large and constant gradient. The orange curve in Figure 10 shows the simulated loss. The pixel–wise error for MAE is given as:(2)$$L_{pix}=\frac{\sum_{i=1}^{n}\left|Y_{i}-Y_{pred_{i}}\right|}{N}$$
where the ground truth data (labeled data) pixel is $Y_{i}$, the estimated depth pixel is $Y_{pred_{i}}$, and the number of pixels in the depth map totals to *N*.

MAE loss is linear and offers equal weight for both lower and higher residual values. This reduces overall prediction and hence, MAE is not good for learning. The reversed Huber loss function (BerHu Loss) [6,9] is a better alternative. This provides higher weight to pixels with higher residuals (mean square error) and also to pixels with smaller residuals similar to MAE (Figure 10, dark green curve). The transition between mean square error (MSE) and MAE is defined by a threshold *c*. In a training batch, *c* is 20% of the maximum error. The BerHu Loss is given as:(3)$$\begin{array}{c} L_{pix}=\left\{\begin{array}{cc} & |Y_{pred_{i}}-Y_{i}\left|\;\;\;\;\;\;\;\;\;\right|Y_{pred_{i}}-Y_{i}|\leq c,\\ & \left({\left(Y_{pred_{i}}-Y_{i}\right)}^{2}+c^{2}\right)/2c\;\left|Y_{pred_{i}}-Y_{i}\right|>c\\ & c=0.2max_{i}\left(\right|Y_{pred_{i}}-Y_{i}\left|)\right\} \end{array}\right. \end{array}$$

The depth map needs parameters to define the image structure and object features other than pixel–wise errors. This information comes from interdependent neighboring pixels [49]. The image structure is constructed by the Structural Similarity Index (*SSIM*). The perceptual difference is indicated by *SSIM* loss between images. Identical images score the lowest. The loss function is:(4)$$L_{SSIM}=1-SSIM\left(Y_{i},Y_{pred_{i}}\right)$$
when *SSIM* loss is 0, the two images have the same structure. Usually, a weight of 0.5 is applied to this loss to weaken the penalization.

The features and edges in an image are represented by high–frequency structure components. Edge functions make the depth map sharper and more detailed. Gradient edge loss functions are based on the maximum of the derivatives. However, this function sometime leads to double edges as in lines. The derivative will give one positive and one negative peak in such cases. The intensity gradient of an image *Y* is given in the horizontal and vertical directions as:(5)$$\begin{array}{c} \frac{\partial Y}{\partial h}=Y\left(h+1,v\right)-Y\left(h-1,v\right),\\ \frac{\partial Y}{\partial v}=Y\left(h,v+1\right)-Y\left(h,v-1\right),\\ L_{edges}=mean(|\frac{\partial Y_{pred}}{\partial v}-\frac{\partial Y}{\partial v}\left|+\right|\frac{\partial Y_{pred}}{\partial h}-\frac{\partial Y}{\partial h}\left|\right) \end{array}$$

We used a comprehensive loss function formulated on the above loss functions. These function outputs are further weighted with hyper–parameters to control the penalization. Our loss function comprises pixel–wise loss (MAE or BerHu), structural loss (*SSIM*), and edge loss (gradient). The total loss is:
(6)$$\begin{array}{cc} L_{total}\left(Y_{i},Y_{pred_{i}}\right)= & \lambda_{1}L_{pix}\left(Y_{i},Y_{pred_{i}}\right)+\lambda_{2}LSSIM\left(Y_{i},Y_{pred_{i}}\right)\\ \; & +\lambda_{3}L_{edges}\left(Y_{i},Y_{pred_{i}}\right) \end{array}$$

## 5. Datasets

The NYU dataset [50] provides registered RGB–depth image pairs of 640 × 480 pixels resolution. The dataset contains diverse 464 indoor scenes. The depth images are obtained from a Kinect 3D camera which is processed (in painted) to fill in the missing information. Hence, the pixel–level depth information is semi–synthetic. These semi–synthetic depth maps have a maximum depth range of 10 m. The dataset is popular for semantic and depth–related studies and is, hence, suitable for comparing performance. This dataset is divided into three parts and used for training models, validating loss, and finally evaluating our models.

## 6. Standard Performance Metrics

Performance evaluation and comparison of our trained models are based on several prior works [5,7,51,52]. These are
Root Mean Squared Error (*RMS*):(7)$$RMS=\sqrt{\frac{1}{N}\sum_{i\in N}{\left(Y_{i}-Y_{pred_{i}}\right)}^{2}}$$Average relative error (*REL*):(8)$$REL=\frac{1}{N}\sum_{i\in N}\frac{|Y_{i}-Y_{pred_{i}}|}{Y}$$Logarithm error ($log_{10}$):(9)$$log_{10}=\frac{1}{N}\sum_{i\in N}\left|log_{10}\left(Y_{i}\right)-log_{10}\left(Y_{pred_{i}}\right)\right|$$


The pixel–level percentage having a relative error below the defined threshold (1.25, 1.25^2^, 1.25^3^) is defined as threshold accuracy ($\delta_{i}$). It is based on the maximum ratio of labeled data pixels and predicted pixels [51]. This is represented as $Y_{i}$% s.t. max($\frac{Y_{pred_{i}}}{Y_{i}}$,$\frac{Y_{i}}{Y_{pred_{i}}}$) = δ < th for th = 1.25, 1.25^2^, 1.25^3^; and *Y* is the average value of pixels in labeled data.

Smaller values of *RMS*, *REL*, $\log_{10}$ error are the goals here while higher values of δ below the defined threshold are a good indicator.

## 7. Experiments and Ablation Studies

We train the model on Google Co Laboratory Pro. This gave us a faster GPU (T4, P100/V100) with 25 GB GPU memory. The training was stopped after 10 epochs so as to compare performances. Batch sizes 4 and 8 were used to meet the allocated memory limits. The learning rate was 0.0001. The learning rate exponentially decayed for successive epochs. The filter weights were randomly initialized. The loss optimizer was ADAM. The Batch Normalization layer in our network reduces internal Covariate Shift, speeds up training, and reduces overfitting. This layer has two learnable parameters (β and γ) and two non–learnable parameters (Mean and Variance Moving Averages). The original paper [53] proposes this layer before the non–linear activation function. However, many researchers advocated better results when this layer is placed after the activation function. Hence, as part of the ablation study, this aspect will be verified. Convolutional layers extract features from input images through learnable filter parameters and remove redundant information by weight sharing. Higher convolution layers lead to an exclusive compressed feature map of the image which ultimately provides informative decisions. This layer also consumes most of the training time. Optimal use of convolution layers is thereby necessary. We experiment with the density of convolution layers in our network architecture and study the performance. The activation layer filters the information (neuron) transmitted to the succeeding layer by a non–linear function. These layers activate the selected neurons by increasing their weight. ReLU (Rectified Linear Unit) is computationally efficient for positive values and allows backpropagation. However, negative input values prevent backpropagation and learning. The Leaky ReLU (LR) activation overcomes this problem by having a small positive slope in the negative zone. ELU (Exponential Linear Units) are better than these two activation functions, but are computationally intensive and, hence, not used. In practice, there is no evidence that Leaky ReLU is always better than ReLU. Hence, we experiment to compare the performance between ReLU and LR. We developed many models by changing the blocks of the architecture of our network to study the impact of the various blocks on the overall performance of (a) the activation layer ReLU and Leaky ReLU; (b) the batch normalization density; (c) the sequence of batch normalization and activation layer: before or after; and (d) the convolution layers in the stack. We also experimented with different loss functions for training. The model implementation and training are in the following combinations:NDWT (3C, 3R, 3Bs) + BsNADWT (3C, 3LR, 1Bs)NADWT (3C, 3LR, 1Bs) + BsNADWT (3C, 3Bs, 3R) + BsNADWT (3C, 3R, 3Bs) + BsNRDWT (3C, 3R, 3Bs) + BsNRDWT (3C, 3Bs, 3R) + BsNARDWT(3C, 3LR, 3Bs) + BsNARDWT (3C, 3R, 3Bs) + BsNARDWT (3C, 3Bs, 3LR) + BsNARDWT (3C, 3Bs, 3LR)NARDWT (3C, 3LR)NARDWT (4C, 4Bs, 4LR) + 1Bs

Where, C: Convolution LAYER, R: ReLU, LR: Leaky ReLU, Bs: Batch Normalization, and NUMBER: Number of LAYERS implemented.

Here, NDWT is our basic model, which implements multistage networks with skip layers. There are five down–sampling blocks (block D in Figure 4) with one input and two outputs. These outputs cater to the lower D block and skip layer (Figure 2). The sequence of the activation layer and bath normalization layer is interchangeable in our studies. The DWT layer replaces the Maxpool layer in this network. These blocks make the encoder. The upsampler block (U as in Figure 7) is similar to the D block with the exception of the DWT layer. An IWT layer at the output upscales the estimated image. There are 10 such blocks which for four decoder chains. The last decoder U15 takes all outputs of decoders via skip paths to provide the final estimate. NADWT augments our base with attention gates (Figure 8) in all the skip functions, which makes the training focus on image zones of higher energy. In the NRDWT model, we replace the convolution with residual convolution as in Figure 5 for encoders and decoders. This model augmented with attention gates gives NARDWT model.

The NYU dataset is used by most researchers and hence we used this for bench–marking. The trained model was evaluated with performance metrics as given by [5,7]. This gives an easy and error–free comparison method. The evaluation dataset (NYU–test) is used. We experiment with the hyper–parameters $\lambda_{1}$ to $\lambda_{3}$ and empirically find that the optimum weights are $\lambda_{1}$ = 0.5, $\lambda_{2}$ = 1 and $\lambda_{3}$ = 0.1. We also modified the loss function and replaced the MAE with BerHu.

The trained models were tested on some complex indoor images having good depths and variation in contrasts. The performances of our models are plotted in Figure 11, Figure 12 and Figure 13. The training loss and training accuracy are given in Figure 14 and Figure 15. The validation loss and validation accuracy are given in Figure 16 and Figure 17.

In Figure 11 we take UNET++ as the base for work (Figure 11C). The details of feature depths are barely visible. Our basic NDWT (Figure 11(1)) has a configuration of three convolution layers, three ReLU layers, and post–batch normalization. This model provides better feature detailing showing that edges and high–frequency features are propagated from early encoder features to the estimated features. This model is augmented with attention gate as NADWT (3C, 3LR, 1Bs) NADWT (3C, 3LR, 1Bs) + Bs, NADWT (3C, 3Bs, 3R) + Bs, and NADWT (3C, 3R, 3Bs) + Bs (Figure 11(2–5)). An attention grid generated in these models improves the areas of relevance with higher weight and brings out the object boundaries (Figure 11(2)). LR activation with batch normalization layers only at the final stages improved the depth dynamic range (Figure 11(3)). This model gives the best performance. The same effect is seen with batch normalization layers before each activation layer (Figure 11(4)). Batch normalization after each activation layer reduces the depth range (Figure 11(5)) as the activation function has pruned the lower value neurons. NDWT with residual convolution further improves the object details (Figure 11(6,7) left corner objects) but blurs the edges lightly. Here, again, batch normalization before the activation layer is better. An attention gate is also added to NRDWT models (Figure 11(8–13)). Here, a model with R activation layers makes the estimation better when compared with ground truth. A reduction in the batch norm before the final output shows a slight loss of detail of the sofa arm (Figure 11(11)). Removing all batch norm layers leads to the degradation of definitions in the estimated image (Figure 11(12)). We increased the convolution layers in one model (Figure 11(13)). The near objects are stronger but definitions at the end of the room are lost. Figure 12 plots the performance of training loss of each of the 13 models. We also show the performance of UNet with DWT here for comparison. NADWT (3C, 3LR, 1Bs) + Bs has the lowest loss and correlates with the depth image in Figure 11(3). The model evaluation accuracy performance (Figure 13) also supports this. Figure 14, Figure 15, Figure 16 and Figure 17 show the model training loss performance, accuracy performances, and validation accuracy performances. The curves are close, indicating that the models are well–trained. The jagged lines are indications of over–fit or under–fit. The training is fast and reaches near saturation within 10 epochs.

## 8. Results and Observation

Figure 11 shows the visual quality of models after training. We summarize our observations below:Batch normalization: improves the depth range and loss. Batch normalization after the activation layer degrades loss. Additional computations and trainable parameters.Activation: among activation layers, the LR activation function offered higher performance. Training and validation performance is better with ReLU.Attention: gives higher training, validation, and evaluation scores.Residual: gives lower training and validation accuracy but the evaluation score is moderately better. Requires more training.Convolution: higher convolution layers do not improve performance, but visually give better representation.Loss function: replacing MAE loss with BerHu loss did not show improvement.

The overall best loss performance is from the NDWT + Attention (NADWT) network followed by the NDWT model and third the Residual (NRDWT) model (Figure 12 and Figure 13). It is also observed that the NARDWT models need more training iterations. The best training accuracy is from the NDWT model followed by the NARDWT model. The validation accuracy performance plots (Figure 17) show that NARDWT models tend to saturate faster. We verified the performance of our model–3 with Haar wavelets instead of db4 wavelets. There seems to be a minor improvement in performance as tabulated in Table 2.

The performance parameters are tabulated in Table 3. Our models are superior to published DWT–type models and UNET++ for depth prediction. Our network has better scores for all six types of performance metrics. All our model variants performed better as seen in Figure 12 and Figure 13. The primary improvement is due to higher convolution layers. We experimented with low convolution layer density NDWT having two layers and NDWT with three layers. The performance was better with three layers. This was the optimal number as blocks with four convolution layers had a lower performance. The performance further improved with the inclusion of the attention function. Attention enables the learning of finer structures leading to performance improvements. Regularization with bath normalization before the activation layer corrects the co–variance shift of learned weights. This adds to the improvement. We also compared our results with a UNET based on a DenseNet backbone encoder. This model performed higher as (1) it used pretrained models and weights (2) it was additionally trained on a more extensive set of improved image sets (50,000) and (3) The trainable parameters are very large. We trained our model from scratch on less than 400 image sets from NYU.

The training time increases with convolution density, batch normalization layers and density, attention feature, and residual convolution blocks. The NARDWT models took approximately 17 mins to train with the NYU dataset per epoch using A100, 40 GB GPU (premium GPU). The training increased to about 97 mins for T4 GPU (standard GPU). This was the same with the Haar wavelet–based model. Our light network trains faster compared to UNET with a DenseNet backbone which, in our experimentation, took about 150 min per epoch.

## 9. Conclusions

We developed a DWT–based dense network model that successfully predicts depth from an image. Our network learns to estimate the wavelet coefficients through loss functions consisting of MAE, SSIM, and gradient functions. We experiment with various variants of our network and demonstrate that the performance is better than UNet and UNet++. The network can train fast with NYU datasets, and the average accuracy reaches more than 92% within 10 epochs. The training time of 17 min per epoch is faster than other models based on dense networks. We completed ablation studies with batch normalization and activation types and infer that evaluation performance is best with Leaky ReLU activation and dense batch normalization. The activation layer before batch normalization provided the best–trained models. We studied the density of convolution layers in our models. More convolution layers in a block increase the trainable feature map density and hence higher performance. Higher–density convolution layers yielded better visual results also. The speed of training is an advantage of our model. This speed is primarily due to lower trainable feature maps compared to dense networks like DenseNet, RESNET, etc. The lower feature maps have the disadvantage of lower accuracy. It is observed that estimations are poor for smooth surfaces at the far end of the scene. These aspects require more analysis and study. Further, the network performance for the outdoors will be studied using the KITTI dataset in our subsequent versions of this work. The scope for future studies is increased blocks in the network with higher trainable parameters and pruning of the non–performing weights in these feature maps. This will be a trade–off study for speed and performance. 

## Figures and Tables

**Figure 1 sensors-23-03066-f001:**
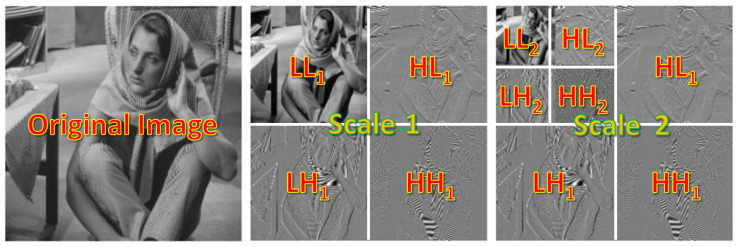
DWT decomposes the image into low–resolution coefficient maps. Here, the scale is 2. We use a scale of 1 to replace the down–sampling operation.

**Figure 2 sensors-23-03066-f002:**
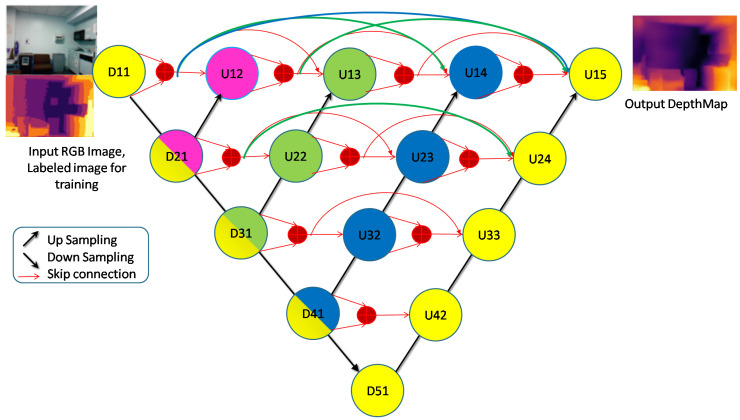
Our network architecture (NDWTN).

**Figure 3 sensors-23-03066-f003:**
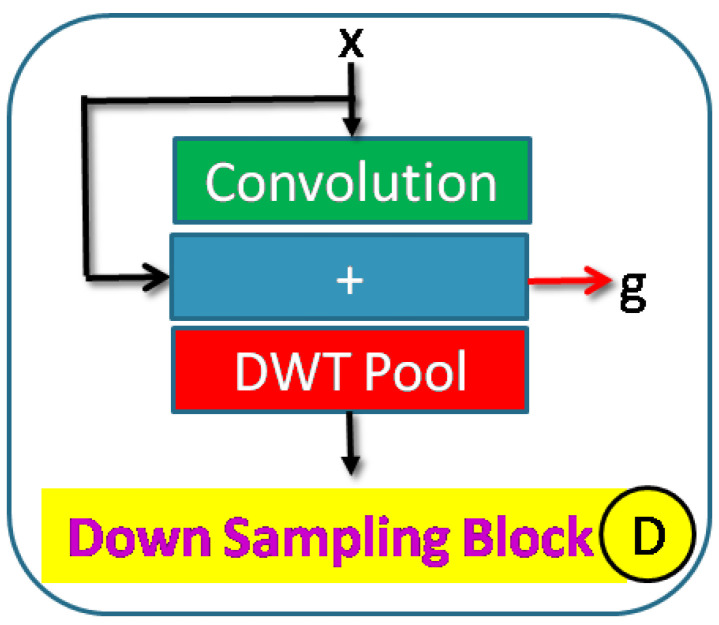
Structure of down–sampling block.

**Figure 4 sensors-23-03066-f004:**
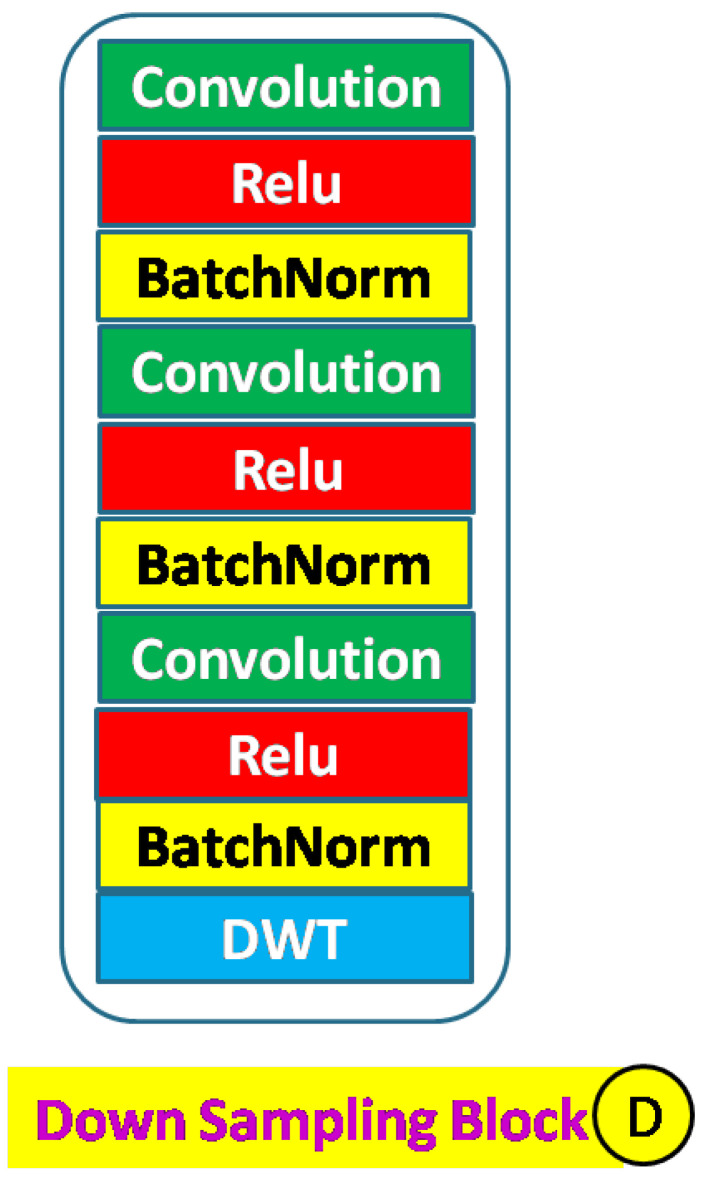
Down–sampling block details: The stack of convolution operators and the sequence of Batch–Norm and activation layers are customized.

**Figure 5 sensors-23-03066-f005:**
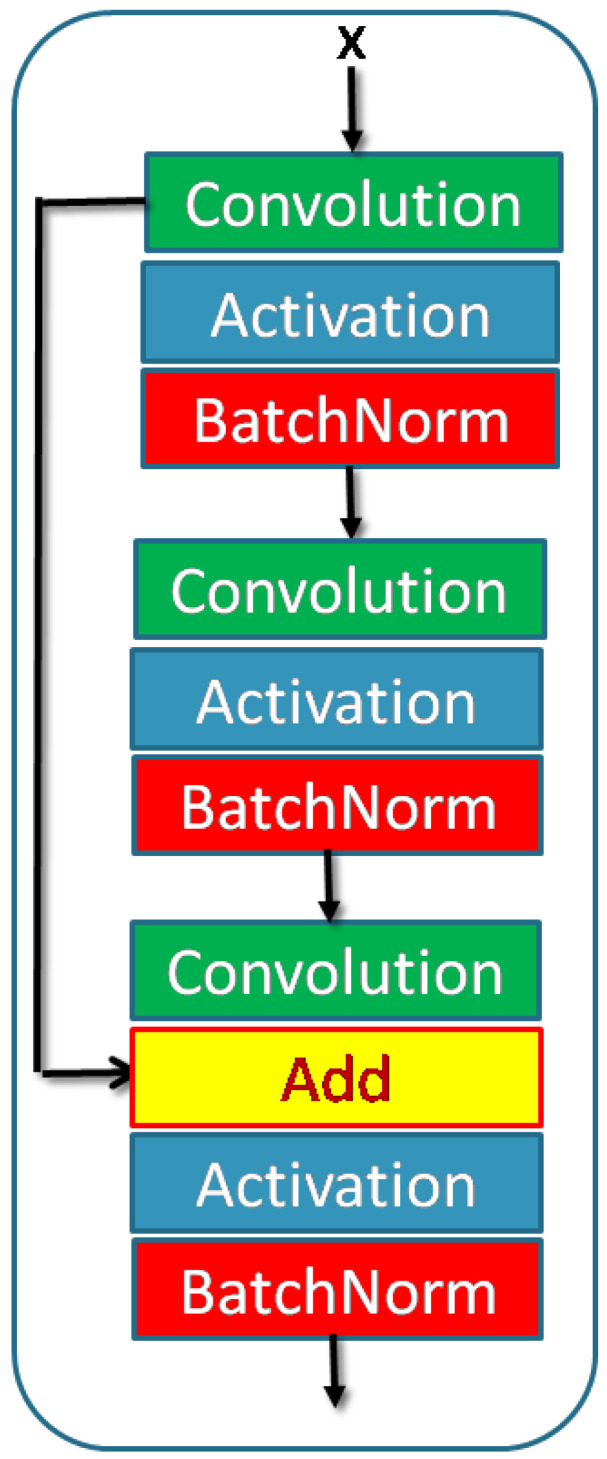
Residual convolution block. The stack of convolution operators and the sequence of Batch–Norm and activation layers are customized.

**Figure 6 sensors-23-03066-f006:**
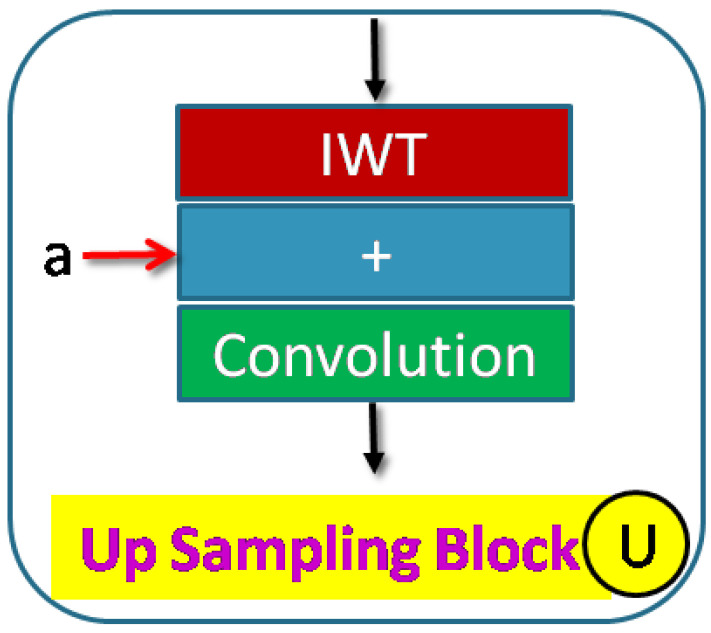
The up–sampling block provides IWT and convolution operations. Information from the skip path ‘a’ is also concatenated.

**Figure 7 sensors-23-03066-f007:**
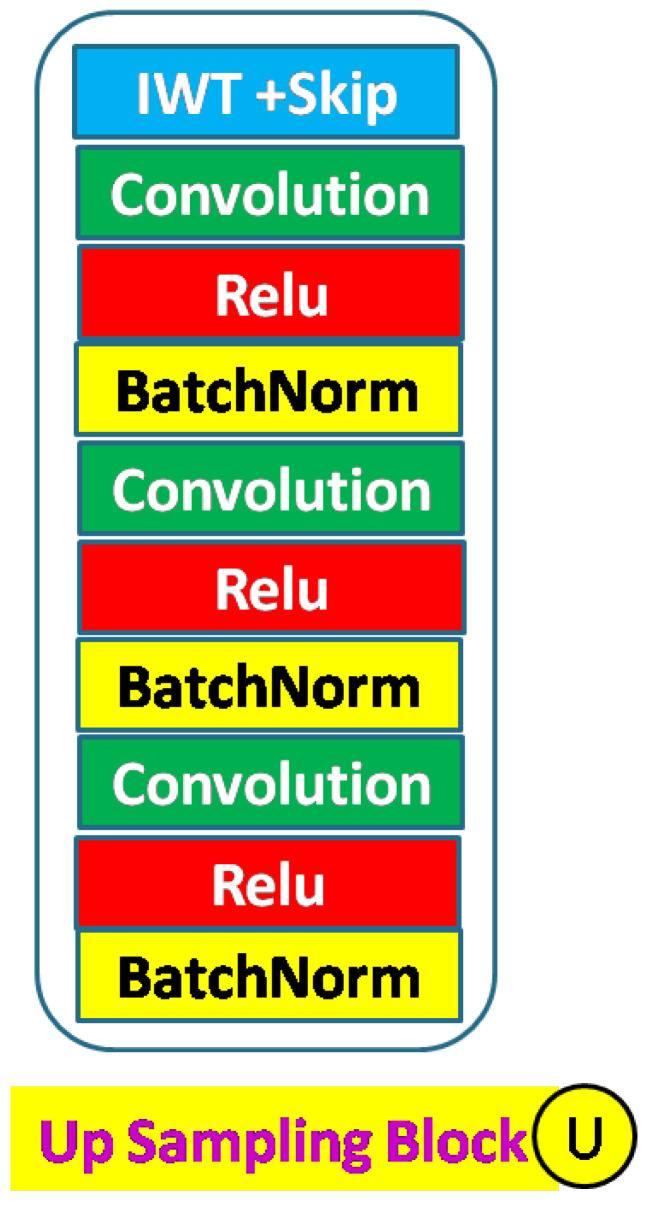
Up–sampling block details: The stack of convolution operators and the sequence of Batch–Norm and activation layers are customized. The convolution stack can be replaced with a residual block.

**Figure 8 sensors-23-03066-f008:**
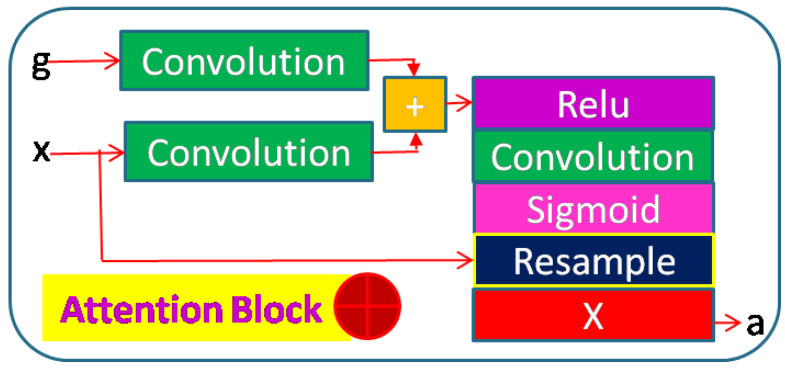
Skip layer with attention. This layer takes two inputs from encoder blocks of different scales, ‘g’ from the higher scale or input to the encoder and ‘x’ from the lower scale or output of the encoder, and feeds the decoder block with attention vectors ‘a’.

**Figure 10 sensors-23-03066-f010:**
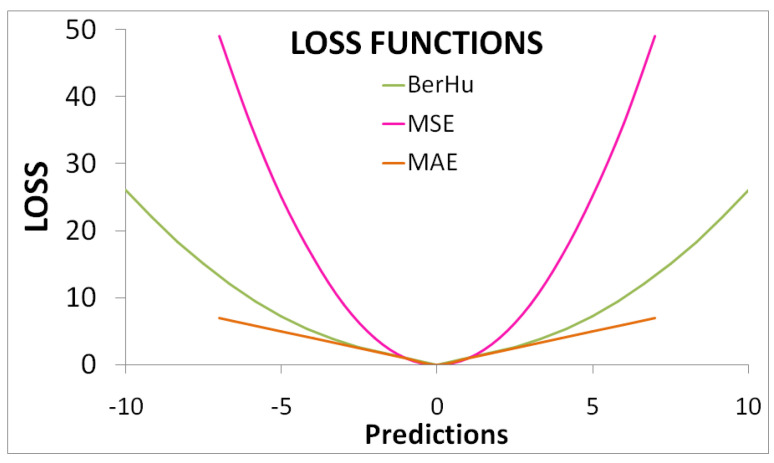
A comparison between MAE, MSE, and BerHu functions.

**Figure 11 sensors-23-03066-f011:**
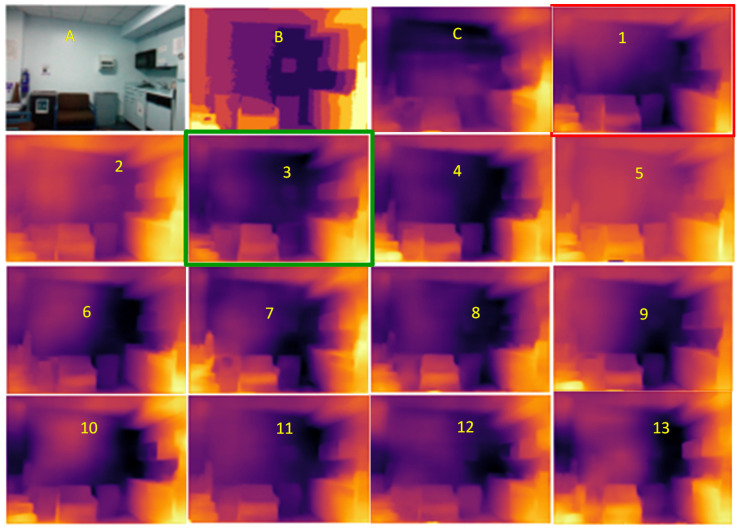
Depth map prediction after training, a visual comparison. **A**: Input image, **B**: Ground Truth, **C**: UNETP, **1**: NDWT (3C, 3R, 3Bs) + Bs, **2**: NADWT (3C, 3LR, 1Bs), **3**: NADWT (3C, 3LR, 1Bs) + Bs, **4**: NADWT (3C, 3Bs, 3R) + Bs, **5**: NADWT (3C, 3R, 3Bs) + Bs, **6**: NRDWT (3C, 3R, 3Bs) + Bs, **7**: NRDWT (3C, 3Bs, 3R) + Bs, **8**: NARDWT(3C, 3LR, 3Bs) + Bs, **9**: NARDWT (3C, 3R, 3Bs) + Bs, **10**: NARDWT (3C, 3Bs, 3LR) + Bs, **11**: NARDWT (3C, 3Bs, 3LR), **12**: NARDWT (3C, 3LR), **13**: NARDWT (4C, 4Bs, 4LR) + 1Bs).

**Figure 12 sensors-23-03066-f012:**
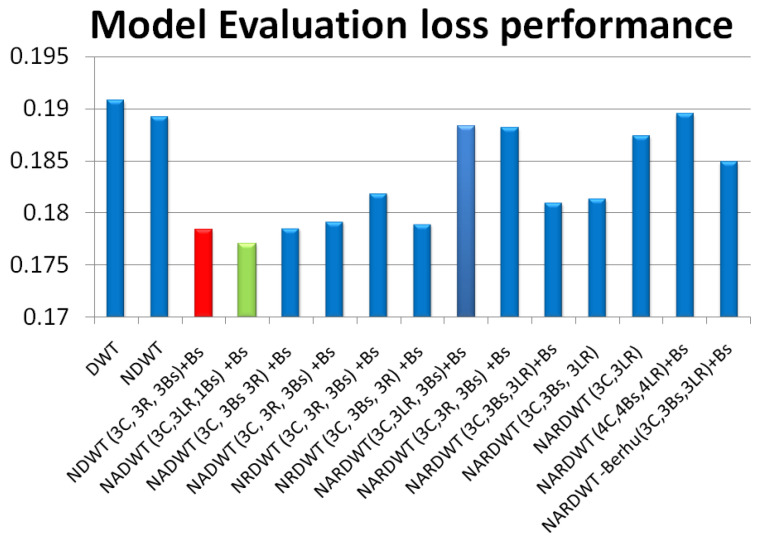
Model loss performance. The best is DWT + Attention followed by Residual + Attention architecture.

**Figure 13 sensors-23-03066-f013:**
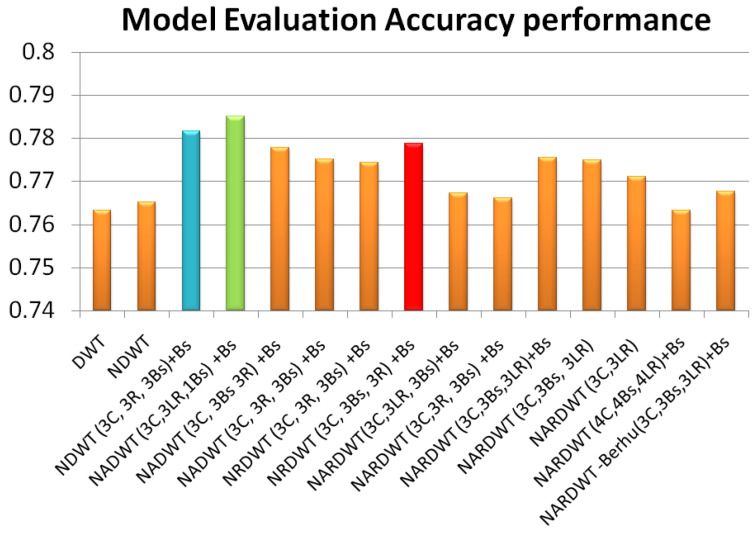
Model evaluation accuracy performance.

**Figure 14 sensors-23-03066-f014:**
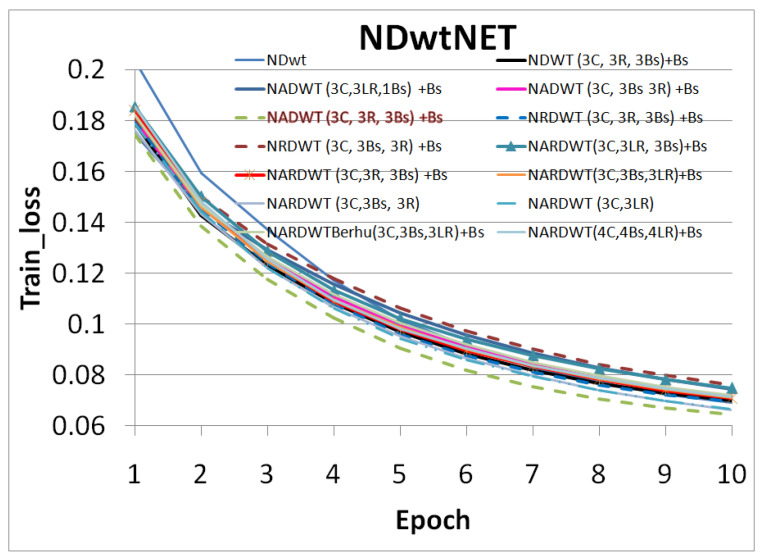
Model training loss performance.

**Figure 15 sensors-23-03066-f015:**
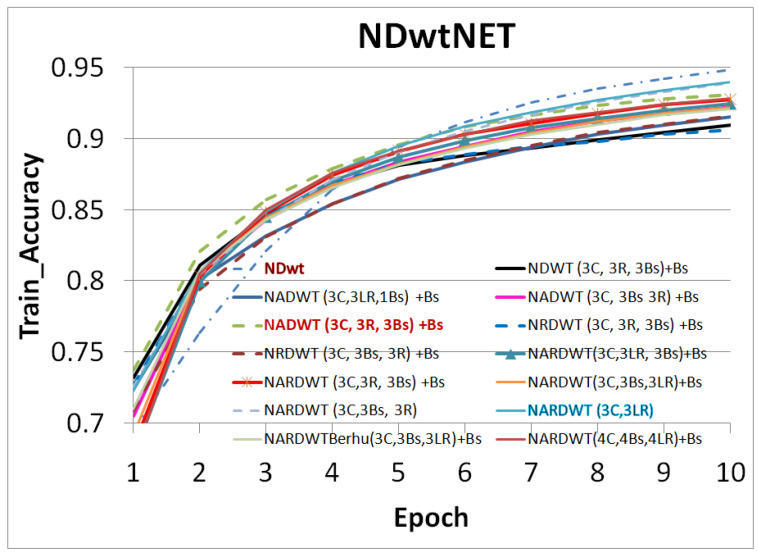
Model training accuracy performance.

**Figure 16 sensors-23-03066-f016:**
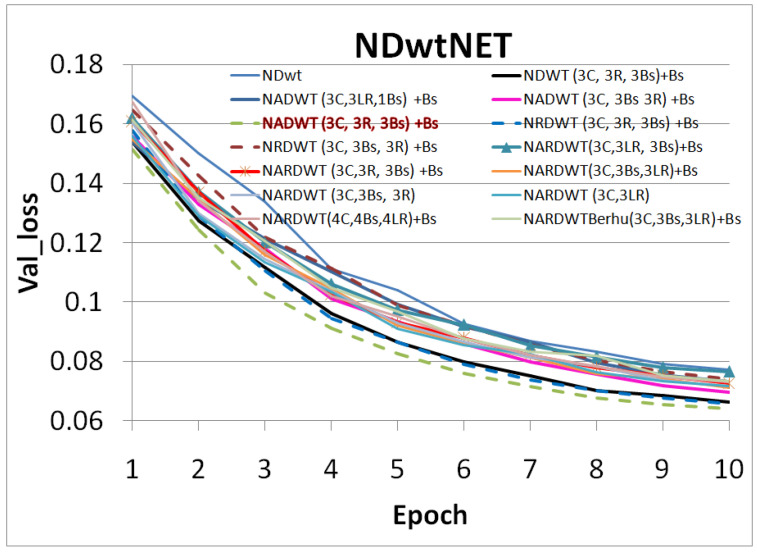
Model validation loss performance.

**Figure 17 sensors-23-03066-f017:**
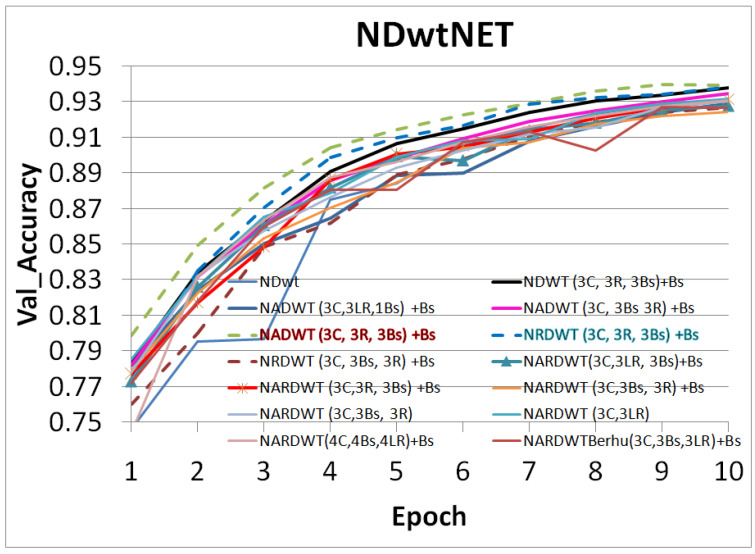
Model validation accuracy performance.

**Table 1 sensors-23-03066-t001:** Parameters and models (in millions).

Parms	DWT	ADWT	UNET++	DenseNet [7]	AdaBins [16]	NDWTN
Total	13.39	14.88	13.23	53.99	78.0	42.82
Trainable	13.39	14.87	13.22	53.97	–	42.66

**Table 2 sensors-23-03066-t002:** Performance of model with different wavelets.

Models	$\delta_{1}↑$	δ2↑	δ3↑	REL↓	RMSE↓	log10↓
db4	0.33	0.61	0.81	0.39	0.16	0.18
Harr	0.34	0.62	0.82	0.39	0.15	0.17

**Table 3 sensors-23-03066-t003:** Comparison of model performances.

Models	$\delta_{1}↑$	δ2↑	δ3↑	REL↓	RMSE↓	log10↓	Year
DWT	0.27	0.52	0.73	0.54	1.76	0.21	2023 *
ADWT	0.27	0.51	0.70	0.80	1.57	0.23	2023 *
UNET++	0.29	0.55	0.75	0.66	1.69	0.21	2023 *
DenseNet [7]	0.85	0.97	0.99	0.12	0.52	0.05	2018
DORN [54]	0.83	0.97	0.99	0.12	0.51	0.05	2018
P3Depth [55]	0.898	0.98	0.996	0.1	0.36	0.04	2022
NewCRFs [56]	0.92	0.99	0.998	0.095	0.33	0.04	2022
ZoeD–M12–N [57]	**0.96**	**0.995**	**0.999**	**0.075**	0.27	**0.03**	2023
NADWT(3)	0.33	0.61	0.81	0.39	**0.16**	0.18	2023

* Trained with NYU dataset.

## Data Availability

Not applicable.

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
