# Peer review of "Nested DWT–Based CNN Architecture for Monocular Depth Estimation"

_sensors, 2023, doi:10.3390/s23063066_

Round 1

Reviewer 1 Report

This manuscript proposes a nested DWT-based CNN architecture for the monocular depth estimation method. See some concerns:

1.    The fourth contributions “An extensive experimental evaluation… the proposed architecture” should be removed. This is not a contribution or novelty.

2.    What are the colors in Figure 2 mean? e.g. D21, D32, what are the half-half colors mean? Similarly, what are the “+” means in Figure 3? There should be an illustration on each figure. Also, such g, x, a in Figure 8 should be described in caption or illustration.

3.    As tabulated in Table 1, the proposed methods is with largest parameters. Why and how do the authors say “The network can train fast with NYU datasets” in the Conclusion?

4.    It would be nice to see a more comprehensive introduction on single-view 3D reconstruction, such as photometric stereo [1, 2, 3]. [1] Photometric method for determining surface orientation from multiple images [2] PS-FCN: A flexible learning framework for photometric stereo [3] Incorporating lambertian priors into surface normals measurement

5.  This manuscript indeed gives an extensive ablation experiment. However, the manuscript lacks compared experiments. The authors almost not compare with state-of-the-art monocular depth estimation methods. Only with several comparisons from baseline architecture and method form arXiv. The authors should add more comparisons to prove the performance of the proposed method.

6.  References are very old, hardly see citations after 2020 – should be updated with an objective view.

7.  As far as I know, the NYU depth dataset is not the only widely used benchmark dataset for monocular depth estimation tasks. Happy to see some results and comparisons on KITTI, which can demonstrate the performance under outdoor conditions.

Some other concerns:

1. There are no compared experiments in Section 3. It should not be a report but have to show the advantages of the proposed method. Even if the authors do not propose algorithms, then the experiment should show the reason why you use them after the comparisons.

2. Lacks some related work of multispectral 3D measurement methods, such as the multispectral photometric stereo method [1,2] for recovering objects' surface shapes.

[1]MPS-Net: Learning to recover surface normal for multispectral photometric stereo

[2]NeuralMPS: Non-Lambertian Multispectral Photometric Stereo via Spectral Reflectance Decomposition

Author Response

We thank the reviewer for investing their valuable time in our paper and providing inputs to improve the paper. We have gone through your feedback and taken suitable action. We have updated our research paper based on your kind suggestions. We have also provided clarifications. All the comments and the corresponding replies/ actions are listed point-wise. We are open to further clarifications.

This manuscript proposes a nested DWT-based CNN architecture for the monocular depth estimation method. See some concerns:

  1. The fourth contributions “An extensive experimental evaluation… the proposed architecture” should be removed. This is not a contribution or novelty.    

Response 1: We agree and have removed the statement. (Updated Lines 107 -108)

  1. What are the colors in Figure 2 mean? e.g. D21, D32, what are the half-half colors mean? Similarly, what are the “+” means in Figure 3? There should be an illustration on each figure. Also, such g, x, a in Figure 8 should be described in caption or illustration.    

Response 2:  Thanks for the feedback.

NDWT has a single encoder path and multiple decoder paths of different scales, all connected through dense skip connections. These skip connections enable nested networks which reduces the semantic gap and provides deep supervision of the output. NDWTN has four scales having a UNet structure. The networks and scales are indicated with yellow, blue, green, and pink colors. Each decoder has independent outputs which are connected to the final output through skip connections. (Updated Lines 174 -180)

Figure.3 shows a generic down-sampling block. In some experiments, the block implemented residual convolution with feedback from the input to the output. The '+' refers to the addition of input and the convolution output.  The detailed part is shown in Figure 5 and the text is updated in Lines 182-183.

In Figure 8, a simplified schematic of the skip layer with attention is shown. This section takes two inputs, one from the preceding encoder block (higher scale) as ‘g’ and one from the horizontal encoder block (same scale) as ‘x’ while a is the output of the attention block. The captions are updated as:

“Figure 8. Skip layer with attention. This layer takes two inputs from encoder blocks of different scales, ‘g’ from the higher scale or input to the encoder and ‘x’ from the lower scale or output of the encoder, and feeds the decoder block with attention vectors ‘a’.” (Updated Fig. 8 caption between Lines 220 -221)

  1. As tabulated in Table 1, the proposed method is with largest parameters. Why and how do the authors say “The network can train fast with NYU datasets” in the Conclusion?    

Response 3: We agree that the training is dependent on the machine performance, datasets, loss functions, and training hyperparameters.  Our NDWTN has more training parameters (Table 1) and the iterations required for training with the NYU dataset are relatively fast when compared only with DenseNet-based UNet [2]. We found experimentally that, with the same training machine, memory, dataset, training loss, and hyper-parameters our models achieve training accuracy of > 90% within 10 epochs (Fig.15) and take an average of 17 mins per epoch. Network [2] needs to be trained for more than 10 epochs and on average consumed 150 min.

  1. It would be nice to see a more comprehensive introduction on single-view 3D reconstruction, such as photometric stereo [1, 2, 3]. [1] Photometric method for determining surface orientation from multiple images [2] PS-FCN: A flexible learning framework for photometric stereo [3] Incorporating lambertian priors into surface normals measurement

Response 4: Thanks for the references. We have updated the introduction to provide a comprehensive survey of depth estimation. (Updated lines 22-102)

  1. This manuscript indeed gives an extensive ablation experiment. However, the manuscript lacks compared experiments. The authors almost not compare with state-of-the-art monocular depth estimation methods. Only with several comparisons from baseline architecture and method form arXiv. The authors should add more comparisons to prove the performance of the proposed method.

Response 5: We have added the latest SOA results in Table 3. For comparisons, we referred to available publications using the NYU dataset. The references are also updated. Research on monocular depth estimations is towards encoder-decoder architecture, loss function, attention methods, metrics, predictions, etc.  Our work is on low-density UNET and DWT-based architecture and we compared it with similar works. (Table 3 updated)

Models

δ1

δ2

δ3

REL

RMSE

log10

Year

DWT

0.27

0.52

0.73

0.54

1.76

0.21

2023*

ADWT

0.27

0.51

0.70

0.80

1.57

0.23

UNET++

0.29

0.55

0.75

0.66

1.69

0.21

SOA

DenseDepth

0.85

0.97

0.99

0.12

0.52

0.05

2018

DORN

0.828

0.965

0.992

0.115

0.051

0.509

2018

P3 Depth

0.898

0.981

0.996

0.104

0.356

0.043

2022

NeWCRFs

0.922

0.992

0.998

0.095

0.334

0.041

2022

ZoeD-M12-N

0.955

0.995

0.999

0.075

0.270

0.032

2023

NDWT(3)

0.33

0.61

0.81

0.39

0.16

0.18

2023

*Trained with NYU dataset

  1. References are very old, hardly see citations after 2020 – should be updated with an objective view.

Response 6: Thanks for your feedback.

We have updated the introduction with recent research trends and updated the references. We had earlier only referred to publications on UNET and DWT-based architecture or similar methods. (Updated lines 22-102 and Table 3)

  1. As far as I know, the NYU depth dataset is not the only widely used benchmark dataset for monocular depth estimation tasks. Happy to see some results and comparisons on KITTI, which can demonstrate the performance under outdoor conditions.

Response 7: Thanks for the suggestion; we will train the model with the KITTI dataset. This will require handling of the database to meet present compatibility, preprocessing of sparse depth images, and splitting the database for training, validation, and testing about 13 models which have different configuration layers as (a) activation layer Relu and Leaky Relu, (b) Batch normalization and density (c) sequence of batch normalization and activation layer: before or after, and (d) convolution layers density in the stack. Further, we also changed (e) wavelet types- Daubechies/ Haar and (f) and various loss functions for the study.  We feel that the time taken for the study will not be feasible within the given submission time and the analysis will be carried out as a separate study in our future work. 

Some other concerns:

  1. There are no compared experiments in Section 3. It should not be a report but have to show the advantages of the proposed method. Even if the authors do not propose algorithms, then the experiment should show the reason why you use them after the comparisons.

Response 1: Thanks for the observation. Section 3 refers to our architecture and the various variants that we studied. The comparisons with UNET and similar networks are in the form of graphs as in Fig.12 to Fig.17. We have also updated Table 1. We have larger trainable parameters and the performance is better than basic UNet and UNET++. (Updated lines 231-233 and Table 1)

Table 1: Parameters and models (in Millions)

#Params

DWT

ADWT

UNET++

DenseNet[2]

Adabins

Ours

Total

13.39

14.88

13.23

53.99

78.0

42.82

Trainable

13.39

14.87

13.22

53.97

--

42.66

  1. Lacks some related work of multispectral 3D measurement methods, such as the multispectral photometric stereo method [1,2] for recovering objects' surface shapes.

[1]MPS-Net: Learning to recover surface normal for multispectral photometric stereo

[2]NeuralMPS: Non-Lambertian Multispectral Photometric Stereo via Spectral Reflectance Decomposition

Response 2: Thanks for the references. We have added the work in the introduction and references. (Updated lines 22-102)

Please see the attachment for the updated paper.

Reviewer 2 Report

The content of this article is not new, and the experimental part is difficult to illustrate the value of this article, and it does need to be deeply revised, however, I am not sure that the author will really be able to modify it as required:

1. It can be seen from Table 3 that the main comparison methods of the proposed method are traditional DWT, U-net++, and DenseNet, and these are relatively old methods, or they are not models specifically used for depth estimation, which does not explain the advanced nature of the proposed method. It is recommended to compare the methods in this paper with the latest or mainstream depth estimation methods, and it is recommended but not limited to the following:

(1) Deep Ordinal Regression Network for Monocular Depth Estimation (CVPR 2018)

(2) Digging Into Self-Supervised Monocular Depth Estimation (ICCV 2019)

(3) Unsupervised Monocular Depth Estimation with Left-Right Consistency (2017)

2. As shown in Figure 2, the network structure of the method in this article is very similar to that of Unet++, but the expression in Figure 2 is not clear enough, especially the meaning of different color icons is not clear.

3. Although the structure of the NDWTN network is described in this paper, it lacks a description of elements related to depth estimation, such as depth decoding, which is recommended to be added.

4. This method uses MAE loss as training loss, why not use SSIM, depth regularization?
5.In the section "Experiments and ablation studies", I suggest a brief description of the content of each diagram.

6.In the section “Conclusions”, I suggest shortcomings of the work be analyzed.

Author Response

We thank the reviewer for investing their valuable time in our paper and providing inputs to improve the paper. We have gone through your feedback and taken suitable action. We have updated our research paper based on your kind suggestions. We have also provided clarifications. All the comments and the corresponding replies/ actions are listed point-wise. We are open to further clarifications.

  1. It can be seen from Table 3 that the main comparison methods of the proposed method are traditional DWT, U-net++, and DenseNet, and these are relatively old methods, or they are not models specifically used for depth estimation, which does not explain the advanced nature of the proposed method. It is recommended to compare the methods in this paper with the latest or mainstream depth estimation methods, and it is recommended but not limited to the following:

(1) Deep Ordinal Regression Network for Monocular Depth Estimation (CVPR 2018)

(2) Digging Into Self-Supervised Monocular Depth Estimation (ICCV 2019)

(3) Unsupervised Monocular Depth Estimation with Left-Right Consistency (2017)

Response 1: Thank you for the references.

We have added the work in the introduction and references (Updated lines 22-102, 231-233). We have also updated Table-3 which compares with recent papers based on the NYU dataset. Papers [2] and [3] used KITTI and Make3D datasets. In the future, we will add performance with KITTI datasets in a subsequent version of this work.

Table 3: Comparison of Model performances

Models

δ1

δ2

δ3

REL

RMSE

log10

Year

DWT

0.27

0.52

0.73

0.54

1.76

0.21

2023*

ADWT

0.27

0.51

0.70

0.80

1.57

0.23

UNET++

0.29

0.55

0.75

0.66

1.69

0.21

SOA

DenseDepth

0.85

0.97

0.99

0.12

0.52

0.05

2018

DORN

0.828

0.965

0.992

0.115

0.051

0.509

2018

P3 Depth

0.898

0.981

0.996

0.104

0.356

0.043

2022

NeWCRFs

0.922

0.992

0.998

0.095

0.334

0.041

2022

ZoeD-M12-N

0.955

0.995

0.999

0.075

0.270

0.032

2023

NDWT(3)

0.33

0.61

0.81

0.39

0.16

0.18

2023

*Trained with NYU dataset

  1. As shown in Figure 2, the network structure of the method in this article is very similar to that of Unet++, but the expression in Figure 2 is not clear enough, especially the meaning of different color icons is not clear.

Response 2: Thank you for your observation.

The NDWT is seemingly similar to Unet++ but differs as we add (1) attention in the skip and (2) replace the Maxpool layer with the DWT layer. NDWT has a single encoder path and multiple decoder paths of different scales, all connected through dense skip connections. These skip connections enable nested networks which reduces the semantic gap and provides deep supervision of the output. NDWTN has four scales having a UNet structure. Each network and scale is indicated as yellow, blue, green, and pink colors. Each decoder has independent outputs which are connected to the final output through skip connections. (Updated Lines 174 -180)

  1. Although the structure of the NDWTN network is described in this paper, it lacks a description of elements related to depth estimation, such as depth decoding, which is recommended to be added.

Response 3: Thanks for pointing out the shortcoming.

The supervised training attempts to predict pixel-wise depth from models by minimizing the regression loss with the help of ground truth images. In the network, the down-sampling and convolution layers of the encoder reduce the information details of these input images. The feature maps arising from initial encoders have higher resolution but lack global information when compared with the final encoders. Our encoder-to-decoder skip connections improve this detail preservation by concatenating high-resolution local features from initial encoders with the global contextual features of decoders. Each skip connection responds to the encoder features energy using a gating signal to generate an attention grid that preserves the higher local features.  (Updated Lines 199 -207)

  1. This method uses MAE loss as training loss, why not use SSIM, depth regularization?

Response 4: We experiment with many loss functions. Our loss function comprises pixel-wise loss (MAE or BerHu), structural loss (SSIM), and edge loss (gradient) as in Equation 6. Loss functions SSIM, MAE, and BerHu provide global structure. Gradient provides edge enhancement and caters to both local and global depth structures. We use a comprehensive loss function formulated on these loss functions. These function outputs are further weighted with hyper-parameters - λ to control the penalization. The total loss is:

Ltotal (Yi ,Ypredi) = λ1Lpix(Yi ,Ypredi) + λ2LSSIM(Yi ,Ypredi)  + λ3Ledges (Yi ,Ypredi)                   (6)

Where

Lpix(Yi, Ypredi) is either based on  Mean Absolute Error (MAE) or BerHu (Reverse Huber)  loss among the ground truth pixel is Yi  and the estimated depth pixel Ypredi.,

LSSIM (Yi, Ypredi) is the Structural Similarity Index (SSIM) and gives the error based on the perceptual difference of the ground truth and estimated pixels, and 

λ3Ledges (Yi, Yprediis the Gradient edge loss functions based on the maximum of the derivatives of images Y and Ypred. The combined loss function leads to depth estimation and regularization.

  1. In the section "Experiments and ablation studies", I suggest a brief description of the content of each diagram.

Response 5: Your suggestion was valuable. We have added a brief of each depth image.

Here, NDWT is our basic model, which implements multistage networks with skip layers. There are five downsampling blocks (block D in Fig.4) with one input and two outputs. These outputs cater to the lower D block and skip layer (Fig.2). The sequence of the activation layer and bath normalization layer is interchangeable in our studies. The DWT layer replaces the Maxpool layer in this network. These blocks make the encoder. The upsampler block (U as in Fig.7) is similar to the D block with the exception of the DWT layer. An IWT layer at the output upscales the estimated image. There are 10 such blocks which for four decoder chains. The last decoder U15 takes all outputs of decoders via skip paths to provide the final estimate. NADWT augments our base with attention gates (Fig. 8) in all the skip functions, which makes the training focus on image zones of higher energy. In the NRDWT model, we replace the convolution with residual convolution as in Fig. 5 for encoders and decoders. This model augmented with attention gates gives NARDWT model. (Updated Lines 339 - 351)

In Fig.11 we take UNET++ as the base for work (Fig.11.C). The details of feature depths are barely visible. Our basic NDWT (Fig.11.1) has a configuration of 3 convolution layers, 3 Relu layers, and post-batch normalization. This model provides better feature detailing showing that edges and high-frequency features are propagated   from early encoder features to the estimated features. This model is augmented with attention gate as NADWT (3C, 3LR, 1Bs)  NADWT (3C,3LR,1Bs) +Bs, NADWT (3C, 3Bs 3R) +Bs, and NADWT (3C, 3R, 3Bs) +Bs (Fig.11.2 to Fig.11.5). Attention grid generated in these models improves the areas of relevance with higher weight and brings out the object boundaries (Fig.11.2). LR activation with Batch normalization layers only at final stages improved the depth dynamic range (Fig. 11.3). This model gives the best performance. The same effect is seen with batch normalization layers before each activation layer (Fig.11.4).  Batch normalization after each activation layer reduces the depth range (Fig.11.5) as the activation function has pruned the lower value neurons.  NDWT with residual convolution further improves the object details (Fig.11.6, Fig.11.7 left corner objects) but blurs the edges lightly.   Here again batch normalization before the activation layer is better. Attention gates are also added to NRDWT models (Fig.11.8 to Fig.11.13). Here model with R activation layers makes the estimation better when compared with ground truth. Reduction of batch norm before the final output shows a slight loss of detail in the sofa arm (Fig.11.11). Removing all batch norm layers leads to the degradation of definitions in the estimated image (11.12). We increased the convolution layers in one model (Fig.11.13). The near objects are stronger but definitions at the end of the room are lost. Fig.12 plots the performance of training loss of each of the 13 models. We also show the performance of UNet with DWT here for comparison. NADWT (3C, 3LR, 1Bs) +Bs has the lowest loss and correlates with the depth image in Figure Fig.11.3. The Model Evaluation Accuracy performance (Fig.13) also supports this. Fig.14 to Fig.17 shows the model training loss performance, accuracy performances, and validation accuracy performances. The curves are close indicating that the models are well-trained. The jagged lines are indications of over-fit or under-fit. The training is fast and reaches near saturation within 10 epochs. (Updated Lines 362 - 380)

  1. In the section “Conclusions”, I suggest shortcomings of the work be analyzed.

Response 6: Thanks for the feedback. We modified the paragraph.

It is observed that estimations are poor for smooth surfaces at the far end of the scene. These aspects require more analysis and study. Further, the network performance for the outdoors will be studied using the KITTI dataset and like in our subsequent versions of this work. (Updated Lines 447 - 450)

The updated paper is submitted as a reference.

Round 2

Reviewer 1 Report

The authors have addressed all my concerns and I think the current version can be accepted.